# Effectiveness of Local Antibiotics for Infection Prevention in Primary Joint Arthroplasty: A Systematic Review and Meta-Analysis

**DOI:** 10.3390/antibiotics14030214

**Published:** 2025-02-20

**Authors:** Chia-Yu Lin, Chiung-Li Liu, Hon-Lok Lo, Shao-Yuan Hao, Wei-Hsuan Huang, Hsiu-Mei Chang, Tun-Chieh Chen

**Affiliations:** 1Department of Pharmacy, Kaohsiung Municipal Ta-Tung Hospital, Kaohsiung Medical University, Kaohsiung 80145, Taiwan; nancy045775@gmail.com (C.-Y.L.); pinkjelly109@gmail.com (C.-L.L.); annhao@gap.kmu.edu.tw (S.-Y.H.); 920652@gmail.com (W.-H.H.); 880504@kmuh.org.tw (H.-M.C.); 2Department of Pharmacy, Kaohsiung Medical University Hospital, Kaohsiung Medical University, Kaohsiung 807377, Taiwan; 3Department of Orthopedics, Kaohsiung Medical University Hospital, Kaohsiung Medical University, Kaohsiung 807377, Taiwan; honlok1021@hotmail.com; 4Department of Pharmacy, Kaohsiung Medical University Gangshan Hospital, Kaohsiung Medical University, Kaohsiung 820111, Taiwan; 5Division of Infectious Diseases, Department of Internal Medicine, Kaohsiung Medical University Hospital, Kaohsiung Medical University, Kaohsiung 807377, Taiwan; 6School of Medicine, College of Medicine, Kaohsiung Medical University, Kaohsiung 807378, Taiwan; 7Center for Tropical Medicine and Infectious Disease Research, Kaohsiung Medical University, Kaohsiung 807378, Taiwan; 8Center for Medical Education and Humanizing Health Professional Education, Kaohsiung Medical University, Kaohsiung 807378, Taiwan

**Keywords:** local administration, antibiotics, infection rate, arthroplasty, systematic review, meta-analysis, meta-regression

## Abstract

**Background/Objectives**: To evaluate the effectiveness of local antibiotic prophylaxis (e.g., powder, bone cement, intraosseous injection) in reducing periprosthetic joint infections (PJI) and deep wound infections in adults undergoing primary joint replacement surgery. **Methods**: A comprehensive search of PubMed, EMBASE, and the Cochrane Library was conducted from inception to 7 September 2024, including randomized controlled trials (RCTs) and cohort studies without language restrictions. Quality assessment was performed using the Cochrane Risk of Bias (RoB) 2.0 for RCTs and the Newcastle–Ottawa Scale (NOS) for cohort studies. Infection rates were extracted and analyzed using risk ratios (RR) or odds ratios (OR) in a random-effects model with Review Manager (RevMan) 5.4. Sensitivity analysis and meta-regression were also performed to validate the study results and possible risk factors. **Results**: Twelve RCTs and 21 cohort studies were included in the review. Vancomycin powder in the cohort studies demonstrated statistical significance (OR = 0.47, 95% confidence interval (CI): 0.36–0.61, I^2^ = 0%), in contrast to the results in the RCTs (RR = 0.37, 95% CI: 0.06–2.47, I^2^ = 49%). Sensitivity analyses confirmed the robustness and statistical significance of these findings. Both the control and treatment groups primarily cultured Gram-positive pathogens, including in the overall data and specifically for vancomycin powder. The evidence for the use of intraosseous injection (RR of RCTs not estimable, I^2^ not applicable; OR of cohorts = 0.12, 95% CI: 0.02–0.63, I^2^ = 0%) and bone cement (RR of RCTs = 0.40, 95% CI: 0.14–1.17, I^2^ = 56%; OR of cohorts = 1.88, 95% CI: 1.25–2.82, I^2^ not applicable) was inconclusive. Smoking was identified as an important risk factor for post-operative infection. Further research, including more robust trials and cohort studies, is needed to confirm these findings. **Conclusions**: Local administration of vancomycin powder appears effective in preventing deep wound infection after arthroplasty.

## 1. Introduction

Periprosthetic joint infection (PJI) and deep wound infection pose significant challenges following arthroplasty. Managing PJI after arthroplasty often requires major revision surgery and prolonged antibiotic treatment [1]. While the rate of deep incisional infection following primary total knee arthroplasty (TKA) is below 1%, a substantially higher rate of 10.4% has been reported for revision TKA, with complications that are costly to manage [2].

To prevent surgical site infection (SSI) after surgery, the administration of parenteral antibiotics pre-operatively is recommended in the guidelines issued in Taiwan, which are based on those from the Centers for Disease Control (CDC). Typically, cefazolin is administered within 1 h (2 h for vancomycin) before the surgical incision. However, the guidelines provide limited information on the local use of antibiotics following arthroplasty [3].

Previous narrative reviews on local antibiotic applications have explored various methods, including powder, bone cement, and intraosseous injection [4,5,6]. Despite a growing number of studies, the current literature on local vancomycin powder shows inconsistent findings [7,8].

However, some meta-analyses include randomized controlled trials (RCTs) only, while non-randomized studies still make up the majority of the literature evidence. Systematic reviews and meta-analyses have also been conducted, but challenges persist due to variations among the included studies and high heterogeneity [9,10,11]. Common pathogens following arthroplasty include Gram-positive bacteria, such as *Staphylococcus aureus* and *Staphylococcus epidermidis* [5,12].

This study aimed to conduct a systematic review and meta-analysis to assess the effectiveness of various types of local antibiotic applications and to identify isolated pathogens, helping clinicians in selecting appropriate empirical antibiotics.

## 2. Results

### 2.1. Study Search and Characteristics

Among the 1587 retrieved articles, 33 studies involving 78,130 participants met the inclusion criteria for the meta-analysis (Figure 1) [7,8,13,14,15,16,17,18,19,20,21,22,23,24,25,26,27,28,29,30,31,32,33,34,35,36,37,38,39,40,41,42,43]. The basic characteristics of the 33 studies are summarized in Appendix A. The studies were published between 1990 and 2024, with the number of participants enrolled in each study ranging from 20 to 22,889. Most studies have focused on participants who received local prophylactic antibiotic administration (e.g., powder, bone cement, intraosseous injection) in primary joint replacement surgery, with vancomycin powder being the most commonly used method.

### 2.2. Risk of Bias Assessment

The Risk of Bias (RoB) 2.0 tool and the Newcastle–Ottawa Scale (NOS) were used to assess the risk of bias in the RCTs and cohort studies, respectively. The robvis package was used to visualize the risk of bias in the RCTs [45,46]. Bias in each domain and overall bias for the RCTs were visually inspected using the traffic plot. The results are summarized in Appendix A. Overall, seven studies (58.3%) were judged as having “some concerns”, four (33.3%) as “high risk”, and one (8.3%) as “low risk”. For cohort studies, the risk of bias in each domain was assessed using the NOS, with overall scores ranging from 7 to 9 (maximum score of 9), as summarized in Appendix A.

A funnel plot was generated for 28 studies, excluding five articles with no events in either the control or treatment groups (Appendix A). The funnel plot shows asymmetry at the bottom, corresponding to studies with small sample sizes (Peters’ test: *p* = 0.1415; Egger’s test: *p* = 0.0014).

### 2.3. Primary Outcome and Subgroup Analysis

Twelve RCTs and 21 cohort studies were analyzed for deep wound infection or PJI (Figure 2 and Figure 3). The results of the RCTs demonstrate that local antibiotic prophylaxis before primary TKA or total hip arthroplasty (THA) might be effective in preventing deep wound infection or PJI (risk ratio (RR) = 0.39, 95% confidence intervals (CI): 0.16–0.96). Consistent results were observed in the cohort studies (odds ratio (OR) = 0.44, 95% CI: 0.27–0.70).

A subgroup analysis of local antibiotic administration methods (powder, intraosseous injection, and bone cement) was performed. First, the RCT results show that administration of vancomycin powder (RR = 0.37, 95% CI: 0.06–2.47, I^2^ = 49%) and bone cement (RR = 0.40, 95% CI: 0.14–1.17, I^2^ = 56%) did not reach statistical significance. However, the results of the cohort studies with vancomycin powder administration (OR = 0.47, 95% CI: 0.36–0.61, I^2^ = 0%) indicate statistical significance. Second, the cohort results of intraosseous injection show statistical significance (OR = 0.12, 95% CI: 0.02–0.63, I^2^ = 0%), whereas the results of the RCTs are not estimable (RR not estimable, I^2^ not applicable). Third, for bone cement, inconclusive results were observed in both the RCTs and cohort studies, with statistical differences and heterogeneity (RR of RCTs = 0.40, 95% CI: 0.14–1.17, I^2^ = 56%; OR of cohorts = 1.88, 95% CI: 1.25–2.82, I^2^ not applicable).

### 2.4. Primary Outcome and Sensitivity Analysis

A sensitivity analysis was performed to assess whether individual studies significantly influenced the overall results within each subgroup. First, in the RCT subgroup for different types of antibiotic administration, heterogeneity was notably reduced after excluding the study by Abuzaiter et al. [7] for vancomycin powder (RR = 0.16, 95% CI: 0.04–0.75, I^2^ = 0%), and the study by Hinarejos et al. [16] for bone cement (RR = 0.26, 95% CI: 0.10–0.68, I^2^ = 5%) (Appendix A).

Second, the results for vancomycin powder were robust in the sensitivity analysis of cohort studies. However, the results of intraosseous administration may be insufficient to make definitive conclusions. Regarding bone cement administration, heterogeneity was also significantly reduced after excluding the study by Namba et al. [37] (RR = 0.45, 95% CI: 0.35–0.59, I^2^ = 2%) (Appendix A).

Third, additional sensitivity analyses were conducted without subgrouping by the administration type. In the RCTs, the heterogeneity was reduced from I^2^ = 49% to 23% after excluding the study by Hinarejos et al. [16] (Appendix A). Similarly, in the cohort studies, excluding the study by Namba et al. [37] reduced the heterogeneity from I^2^ = 65% to 2% (Appendix A). When combining the results from both the RCTs and cohort studies, the exclusion of both Namba et al.’s [37] and Hinarejos et al.’s [16] studies further reduced the heterogeneity (Appendix A). These findings are consistent with the trends observed in the subgroup analyses of the RCTs and cohort studies (Appendix A).

Fourth, publication bias was assessed using a funnel plot analysis based solely on the sensitivity data from vancomycin powder administration. The funnel plot, along with Peters’ test and Egger’s test, did not show statistical significance (Peters’ test: *p* = 0.7487; Egger’s test: *p* = 0.113) (Appendix A).

Finally, we also conducted meta-analyses of vancomycin powder with different dosage subgroups (Appendix A). Dosing information and the administration timing of the included studies are recorded in Appendix A. In the RCTs, insufficient data were available for the meta-analysis (Appendix A). For the cohort studies, vancomycin powder doses of 1 g (OR = 0.34, 95% CI: 0.21–0.55, I^2^ = 0%) and 2 g (OR = 0.56, 95% CI: 0.41–0.78, I^2^ = 0%) significantly reduced PJI or deep wound infection (Appendix A). An additional sensitivity analysis combining the RCTs and cohort studies was also performed. Vancomycin powder doses of 1 g (OR = 0.36, 95% CI: 0.21–0.61, I^2^ = 17%; excluding Abuzaiter et al. [7]: OR = 0.33, 95% CI: 0.21–0.53, I^2^ = 0%) and 2 g (OR = 0.56, 95% CI: 0.41–0.77, I^2^ = 0%) were consistent with previous data (Appendix A).

### 2.5. Meta-Regression

Meta-regression was performed using R meta packages to identify specific risk factors and evaluate heterogeneity. As reported by Lucenti et al., several variables were discussed in their review, including age, gender, body mass index (BMI), diabetes mellitus (DM), rheumatoid arthritis (RA), and smoking [47]. These variables were incorporated into the meta-regression analysis. Although Lucenti et al. did not mention local antibiotic administration types or study designs, these factors were also included to assess the robustness of the results [47].

The results of the meta-regression are summarized in Appendix A. Most variables examined in the meta-regression did not show statistical significance. Powder administration almost reached statistical significance (*p* = 0.059) when examining the administration types after excluding intraosseous injection and the study by Abuzaiter et al. [7]. Furthermore, the variable “smoking_difference” was identified as a potential risk factor for local antibiotic administration with statistical significance. However, additional studies may be needed to stabilize the meta-regression model for smoking analysis.

### 2.6. Secondary Outcome

The microbiological profile of infected patients was also analyzed, and all reported cultures were recorded. Polymicrobial infections were categorized as “others” due to the lack of specific organism identification. If the specific class, name, or identifier of a pathogenic microbe was not reported, the data were excluded from the analysis. A pie chart was created to display the relative proportions of each type of microbe isolated from the infections. The bacteria were categorized as follows:*Staphylococcus aureus* (*S. aureus*): This includes both methicillin-susceptible *Staphylococcus aureus* and strains where the susceptibility was not reported.Methicillin-resistant *Staphylococcus aureus* (MRSA).Vancomycin-resistant *Enterococci* (VRE).Other Gram-positive bacteria (OGP): This group includes Gram-positive bacteria excluding *S. aureus* or VRE. Some studies may not distinguish between *S. aureus* and VRE, potentially including these bacteria in OGP.Gram-negative bacteria (GNB).Other pathogens: This includes pathogens categorized as “other” in the literature, such as *Mycobacterium abscessus*.

In the control group, other Gram-positive pathogens—including *Streptococcus* and coagulase-negative *Staphylococci* (CoNS)—were the most commonly reported pathogens, accounting for 42.6% of infections. *S. aureus* was the second most common pathogen at 33.4%, with 27.5% being methicillin-susceptible *S. aureus* (or susceptibility not reported) and 5.9% being MRSA. No cases of VRE were reported. A similar trend was observed in the treatment group, with other Gram-positive bacteria accounting for 19.3% of infections, methicillin-susceptible *S. aureus* for 38.6%, and MRSA for 7.0%. As with the control group, no VRE isolates were recorded. Similar trends with vancomycin powder data were observed in both the control and treatment group (Figure 4 and Figure 5). Isolated Gram-positive aerobic pathogens from all local antibiotics or vancomycin powder are also listed in Appendix A.

## 3. Discussion

Our meta-analysis of 21 articles demonstrated that vancomycin powder effectively reduced the risk of deep wound infections or PJI in primary joint arthroplasty, consistent with findings by Gao et al. and Martin et al. [9,10]. Given the similarity of the included data, we also assessed the accuracy of these studies. However, some differences remained among these meta-analyses. First, we separated the results from the RCTs and cohort results due to differences in study design. Sensitivity analyses were also conducted to evaluate the robustness of our findings. Second, studies involving lavage, irrigation, revision surgery, and antibiotic-loaded beads were excluded to reduce the heterogeneity and improve comparability. Excluding these studies, which may have higher infection rates, helped simplify the pooled data. Unlike previous meta-analyses and systematic reviews that included data from revision TKA or THA, we focused exclusively on primary TKA or THA. Studies that included revision operations were only considered if revision-specific data could be excluded.

We analyzed various local antibiotic administration methods to identify the most effective in preventing deep wound infections or PJI in primary TKA or THA. Additionally, vancomycin powder, intraosseous injection and bone cement were used as subgroups in the meta-analysis. Local antibiotic administration may help prevent PJI by achieving high antibiotic concentrations within the wound. Johnson et al. measured both serum and wound vancomycin levels after local administration of vancomycin powder. Their findings demonstrated effective intrawound concentrations with relatively low serum levels, which contributed to preventing infections [48].

Meta-analyses have specifically examined intraosseous injection, including those by Yu et al. and Viswanathan et al. [49,50]. The current data suggest that intraosseous injection shows promise in preventing deep wound infection or PJI. However, due to small sample sizes and the limited number of studies, further research is needed to validate these findings.

Our analysis of bone cement administration revealed greater heterogeneity compared to vancomycin powder and intraosseous injection, likely due to variations in the antibiotics used across the studies (Appendix A). Additional studies are needed to consolidate the findings and strengthen the evidence.

To assess whether the current sample sizes for pooling vancomycin powder provide adequate statistical power, we calculated the optimal information size (OIS) for meta-analysis. The sample sizes in the RCTs and cohort studies for the control and treatment groups met the requirements for sufficient statistical power. Based on the vancomycin powder pooling data from Figure 2 and Figure 3, and assuming a default for α = 0.05 and power = 1 − β = 0.8, the required sample sizes were at least 1185 for the RCTs and 2608 for the cohort studies in each group. These results suggest that the sample size for the RCTs may be insufficient to make a solid conclusion, whereas the sample size for the cohort studies appears sufficient to make robust conclusions.

To assess the robustness of the meta-analysis results, sensitivity analyses were also performed by sequentially excluding individual studies. Exclusion of the studies by Abuzaiter et al. [7], Hinarejos et al. [16], and Namba et al. [37] resulted in a significant reduction in heterogeneity.

Exclusion of Abuzaiter et al. [7]: Excluding this study significantly reduced the heterogeneity, likely due to differences in the antibiotic regimens between the control and treatment groups. The control group was administered with pre- and post-operative antibiotics, whereas the treatment group received only pre-operative antibiotics and topical vancomycin powder (without post-operative antibiotics). The RCT was halted after 1 year due to a higher infection rate in the treatment group. Additionally, the follow-up period of 42 days was shorter than the recommended minimum of three months for assessing deep wound infections or PJI.Exclusion of Hinarejos et al. [16] and Namba et al. [37]: Removing these studies also reduced the heterogeneity. In these studies on antibiotic-impregnated bone cement, variability in the antibiotic types (unreported in Namba et al. [37]) and differences in the stability of the cement-antibiotic mixture may have contributed to the heterogeneity. Further studies are needed to evaluate the use of antibiotic-impregnated bone cement.Exclusion of Erken et al. [31]: In the cohort studies involving vancomycin powder administration, Erken et al. was excluded due to unclear definitions of infection and the follow-up period. After excluding the data from Erken et al., the results of the meta-analysis remained robust (OR = 0.47, 95% CI: 0.31–0.61, I^2^ = 0).

Some articles included populations at high risk for infection, prompting a meta-regression analysis to identify possible risk factors. Risk factors were reviewed based on the variables outlined by Lucenti et al. [47]. Initially, data from the RCTs and cohort studies were pooled using a random-effects model to calculate the RR. The pooling results show that local vancomycin powder administration significantly reduced infection risk (RR = 0.46, 95% CI: 0.34–0.61, I^2^ = 8%). As part of the sensitivity analysis, data from Abuzaiter et al. [7] and Erken et al. [31] were excluded. The results remain statistically significant, with reduced heterogeneity (RR = 0.46, 95% CI: 0.36–0.59, I^2^ = 0%).

According to Lucenti et al., all common risk factors were analyzed for meta-regression under different conditions (overall data and powder administration only) [47]. Smoking status showed significant differences between the control and treatment groups, suggesting its potential impact as a risk factor. Liu et al. conducted a systematic review analyzing the effects of pre-operative smoking and smoking cessation on wound healing and infection in post-operative subjects [51]. Their findings revealed that smoking cessation or being a non-smoker significantly reduced the risk of post-operative wound healing problems (OR = 0.59; 95% CI: 0.43–0.82, *p* < 0.001) and surgical site infections (OR = 0.74; 95% CI: 0.63–0.87, *p* < 0.001) compared to smokers. Although only six studies were included in the meta-regression for smoking analysis, the results suggest that smoking may be influential. Further studies are needed to strengthen this association and improve the meta-regression model. Other risk factors, such as DM, RA, and BMI, were also considered in the meta-regression analysis (Appendix A). Although other metabolic diseases evaluated in the meta-regression did not reach statistical significance, they remain important factors for pre-operative evaluation.

As for powder administration, the results of the meta-regression showed borderline statistical significance after excluding intraosseous injection and the study by Abuzaiter et al. [7]. However, the overall data did not demonstrate statistical significance when comparing intraosseous injection and bone cement administration. This lack of significance may be attributed to the differences between the powder and bone cement administration methods. Further studies are needed to strengthen the meta-regression analysis and clarify these findings.

Regarding the different intervention dosages used in the included studies, both 1 g and 2 g of vancomycin powder may help reduce the risk of PJI or deep wound infection. Studies using a dosage range of 1–2 g may also be effective. However, as only one study investigated the use of vancomycin at 0.5 g, the evidence is insufficient to draw strong conclusions or provide solid recommendations. As shown in Appendix A, there was significant variation in administration timing and methods across the studies. Further research is needed to confirm the effectiveness of fixed dosages with consistent administration timing.

Another potential issue to address is antibiotic resistance. Regarding resistance to vancomycin, we specifically reviewed the records of Gram-positive aerobic pathogens in the included studies to determine if any resistant pathogens were reported (Figure 4 and Figure 5, Appendix A). No cases of VRE were observed in either the control or intervention groups, regardless of whether local antibiotics or vancomycin powder alone were used. Additionally, a similar systematic review reported comparable findings [5]. However, further rigorous studies or antibiotic stewardship programs are needed to evaluate the epidemiology, antibiogram, and antibiotic resistance rates, which are crucial to support the continued use of local antibiotic administration.

There are some strengths and advantages to this meta-analysis. First, it comprehensively evaluated various local antibiotic administration methods. Second, sensitivity analyses and meta-regression were performed to examine the influence of different risk factors and variables. Third, data pooling was performed with similar baselines, and consultations with orthopedic experts were undertaken to enhance the applicability of the study results.

This meta-analysis has several limitations. First, the number of studies on intraosseous injection and bone cement was insufficient to draw solid conclusions. Second, further data are required to refine the meta-regression model for evaluating specific variables and risk factors. Third, the clinical application of cementless THA was not addressed in our meta-analysis. When considering cementless THA or cementation, factors such as bone stability and quality should be taken into account [52]. A previous meta-analysis comparing post-operative infection rates between cementless THA and cementation demonstrated that cementless THA was associated with lower PJI risks [53]. To prevent PJI in cementless THA, pre-operative intravenous antibiotics or novel surface coatings for implants should be considered. Finally, the analysis focused solely on specific orthopedic surgery types, leaving room for further research to expand and analyze the findings for broader clinical applications.

Future research should focus on addressing antibiotic resistance, conducting drug susceptibility tests and offering clinical guidance. Additionally, further evidence is needed to confirm the effectiveness of antibiotic-impregnated bone cement and intraosseous injection. In clinical practice, pre-operative evaluation is crucial to identify resistant pathogens or risk factors associated with poor wound healing. For cementless THA, the application of vancomycin powder should be adjusted based on clinical situations. Therefore, the timing and dosage of administration are important factors to ensure standardized implementation.

## 4. Materials and Methods

This systematic review and meta-analysis followed the 2020 PRISMA guidelines [44]. The protocol was registered before the search (PROSPERO Identifier: CRD42023481792). This study was approved by the Institutional Ethics Committee of Kaohsiung Medical University Hospital (approval no. KMUHIRB-EXEMPT(I)-20240013) and supported by grants from Kaohsiung Municipal Ta-Tung Hospital (grant number kmtth-112-025).

### 4.1. Database Search

Two authors independently conducted a literature search in the PubMed, EMBASE, and Cochrane Library databases up to 7 September 2024, using medical subject headings and free terms to capture the keywords “arthroplasty”, “antibiotics”, “local”, and “prophylaxis”. No restrictions were applied regarding publication language or year. Details of the search strategy are provided in the Appendix A. Additionally, reference lists of the selected studies and reviews were manually searched for further relevant studies. Disagreements between the two reviewers (C.-Y.L. and C.-L.L.) were resolved through discussion, and if disagreements persisted, a third reviewer (T.-C.C.) acted as an arbiter.

### 4.2. Study Selection and Eligibility

We included RCTs and observational cohort studies that compared the outcomes in adults undergoing primary arthroplasties with and without local antibiotic therapy. Studies were excluded if they involved revision surgeries, prior infections, lacked specific infection rate data, or were biomechanical (e.g., in vitro studies), narrative reviews, conference abstracts, study protocols, or case reports. Data from revision surgeries were excluded due to differences in population, including clinical severity, pathogens, and surgical techniques, which may vary from those in primary arthroplasties with PJI.

### 4.3. Data Extraction

The primary outcome was the rate of PJI or deep wound infection, and the secondary outcome was the rate of cultured pathogens in the control and treatment groups. Two authors (C.-Y.L. and C.-L.L.) independently extracted data, including the first author, publication year, number of patients, study design, patient demographics (e.g., age and sex), dosage of local antibiotics, timing of administration, and study purpose. A third author (T.-C.C. or H.-L.L.) acted as an arbiter in cases of disagreement.

### 4.4. Risk of Bias Assessment

The RoB 2.0 tool and the NOS were used to assess the risk of bias in the randomized controlled trials and cohort studies, respectively [45,46]. For RoB 2.0, five domains—randomization process, intended intervention, missing outcome data, outcome measurement, and selection of the reported results—were used to assess bias in the studies reporting infection rates. For NOS, three domains—selection, comparability, and outcome—were applied to evaluate bias in the cohort studies on infection rates. Two researchers (C.-Y.L. and either C.-L.L. or S.-Y.H.) independently assessed the bias by answering signal questions within each domain, using a hypothetical randomized trial as a reference. A third author mediated any disagreements until consensus was reached.

### 4.5. Statistical Analysis

For infection rate and subgroup comparisons, we calculated the pooled RR or OR with a 95% CI using a random-effects model, applied separately to the RCTs and cohort studies. Due to methodological differences between the RCTs and cohort studies, the results were pooled separately by study type [54]. Statistical heterogeneity was evaluated with the I^2^ value, the Cochrane χ^2^-test (Q-test), and *p* < 0.05 as the threshold for statistical significance. Sensitivity analysis was conducted by sequentially excluding each study to assess the impact of individual studies on the overall pooled estimates or by data pooling the different study types within the same subgroup. The extracted data were analyzed using RevMan 5.4. Publication bias was assessed through Peters’ test, Egger’s test, and visual inspection of the funnel plot using R version 4.4.1 with the ‘meta’ package [55].

Meta-regression was performed using a mixed-effects model to explore potential sources of heterogeneity and identify the risk factors influencing infection rates. The variables analyzed individually included the type of local antibiotics, study design, gender (male), age, BMI, smoking status, and diagnoses of DM and RA. The meta-regression was conducted using R version 4.4.1 with the ‘meta’ package [55].

## 5. Conclusions

Local administration of vancomycin powder appears effective in preventing deep wound infection following arthroplasty. Additional trials and cohort studies are needed to validate the results for intraosseous injection and bone cement. Standardization of local antibiotic administration, including location and dosage, is crucial to support this strategy. The primary pathogens identified were Gram-positive bacteria, particularly *Staphylococcus aureus*. Smoking was identified as an important risk factor for post-operative infection. Further research, including larger and more robust trials, is necessary to confirm these findings.

## Figures and Tables

**Figure 1 antibiotics-14-00214-f001:**
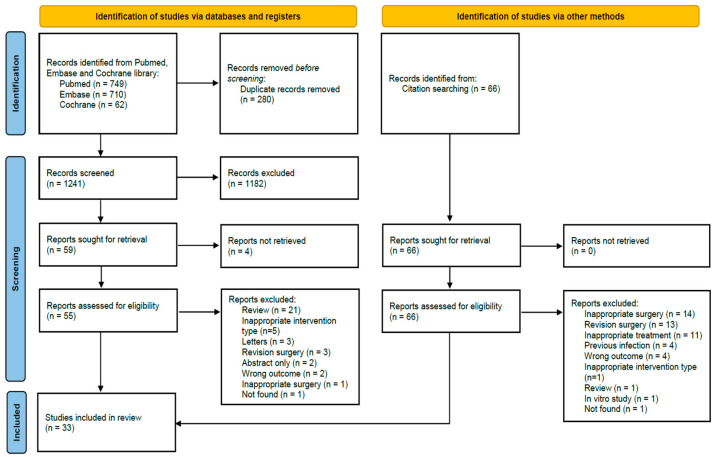
Preferred Reporting Items for Systematic reviews and Meta-Analyses (PRISMA) 2020 flow diagram for study inclusion [44].

**Figure 2 antibiotics-14-00214-f002:**
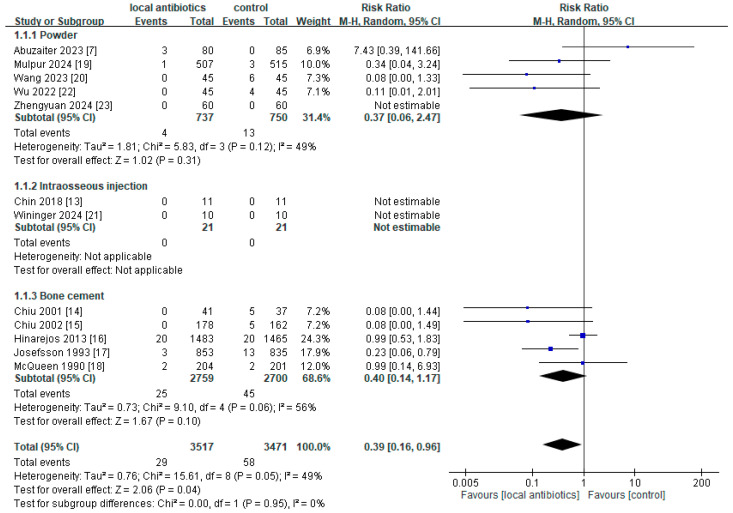
Forest plot of RCTs with subgroup analysis by route of administration.

**Figure 3 antibiotics-14-00214-f003:**
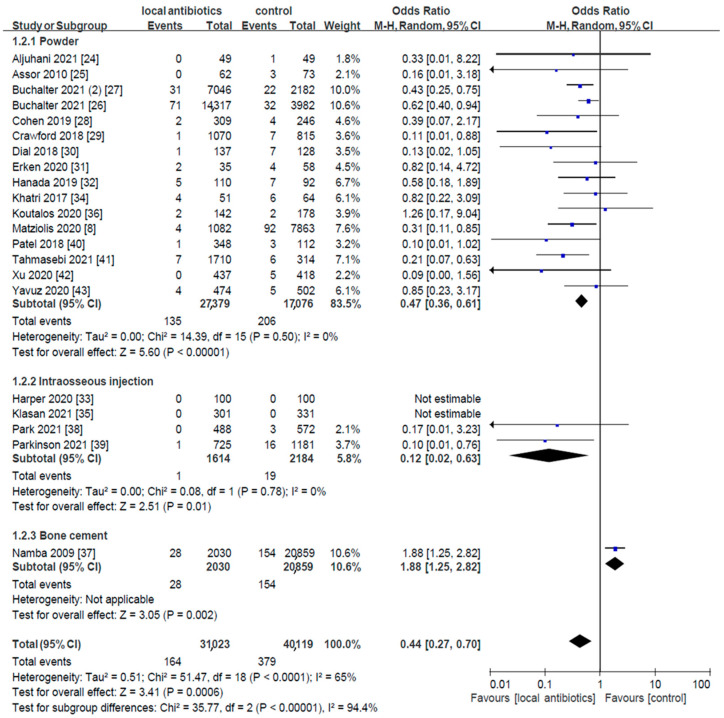
Forest plot of cohort studies with subgroup analysis by route of administration.

**Figure 4 antibiotics-14-00214-f004:**
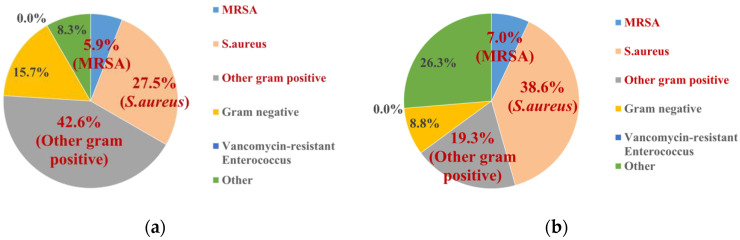
Isolated pathogens from overall data: (**a**) control group; (**b**) treatment group. No VRE isolates (0%) were recorded in the control and treatment group.

**Figure 5 antibiotics-14-00214-f005:**
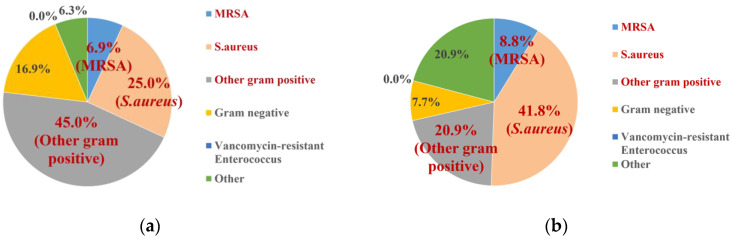
Isolated pathogens from vancomycin powder data: (**a**) control group; (**b**) treatment group. No VRE isolates (0%) were recorded in the control and treatment group.

## Data Availability

Data are available upon request from the corresponding author.

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
