# Peer review of "Effectiveness of Local Antibiotics for Infection Prevention in Primary Joint Arthroplasty: A Systematic Review and Meta-Analysis"

_antibiotics, 2025, doi:10.3390/antibiotics14030214_

Round 1

Reviewer 1 Report

Comments and Suggestions for Authors

Thank you for this interesting paper, about an important topic in orthopedic surgery. With increasing numbers of arthroplasties infection prevention is a hot topic. However, I have some suggestions.

Title:

Concisely describes the research focus, and Includes key terms that highlight its relevance.

Abstract:

Short and on-point, provides a clear background, objective, methodology, and results. Maybe include more quantitative results (e.g., confidence intervals for bone cement or intraosseous injection outcomes).

Introduction:

Clearly states the problem of periprosthetic joint infections (PJIs) and the need for local antibiotic prophylaxis. Relevant literature to justify the study is cited.

The rationale for separating randomized controlled trials (RCTs) and cohort studies could be better justified.

Materials and Methods:

Adheres to PRISMA guidelines and provides a PROSPERO registration. Search strategy, inclusion/exclusion criteria, and quality assessment tools are described properly.

The exclusion of studies involving revision surgeries is mentioned but not justified in detail.

Furthermore, no information on how inter-reviewer disagreements were resolved is mentioned. Variability in reported antibiotic dosages is not addressed. Discuss how dosage variability impacted conclusions and suggest standardization for future research

Results:

Includes robust meta-analysis with subgroup and sensitivity analyses – results are presented in a structured manner.

High heterogeneity for bone cement and intraosseous injection results is insufficiently explained.

Discussion:

Compares findings with previous meta-analyses and systematic reviews and highlights vancomycin powder as effective in cohort studies. However, implications of inconclusive results for bone cement and intraosseous injection are not mentioned at all. Include potential mechanisms (e.g., localized drug concentrations) for vancomycin powder's efficacy.

Conclusion

Key findings are well summarized.

General:

This manuscript presents valuable insights but requires significant minor revisions to ensure the quality needed to be published in your journal.

sincerely

Reviewer 2 Report

Comments and Suggestions for Authors

The authos have performed a review study of thr Effectiveness of Local Antibiotics for Infection Prevention in Primary Joint Arthroplasty: A Systematic Review and Meta-Analysis

The following comments are provided to improve the manuscript and increase the potential number of readers.

The authors should clarify which conclusion is supported by the data. Are they advocating the use of vancomycin powder (details of administration not specified) for all patients subject to TKA and THA? If so, what about the risk of developing widesperad antibiotic resistance? 

Looking at the risk factors, only smoking would be considered for possible patients' risk stratification and not diabetes or other metabolic diseases?

The current trend in THA operation is cementless. How to prevent PJI with antibiotic laced bone cement if it is not used?

Are the authors suggesting intraosseous antibiotic injection in both femoral and tibial canal for TKA and acetabular and femoral bone for THA?

The authors cannot limit themselves only to report the statistical analysis but need to discuss and suggest some guidelines deriveed from the study of the literature versus suggesting that further studies are needed without specifying which ones would be necessary to confirm the findings. Otherwise, the manuscript as the authors also report, would in large part only confirm previous data without a significant addition to the literature. 

In summary, it is expected a critical analysis of the literature, the gaps which need to be filled in this field of research, and suggested measures based on the analysis to effectively reduce PJi (and deep wound infection) without increasing antibiotic resistance.

Round 2

Reviewer 2 Report

Comments and Suggestions for Authors

English language should be reviewed due to some inaccuracies or difficult understanding of the meaning of some sentences. One example is below.

Lines 213-217. The possible reason why local antibiotic administration can prevent PJI might be related to intra-wound high concentration. According to Johnson et al., vancomycin powder local administration may produce highly therapeutic intra-wound concentrations while yielding low systemic levels to prevent infections

Comment: Suggested change. The possible reason why the local administration of an antibiotic agent may be effective in preventing PJI is related to intra-wound high concentration. According to Johnson et al., the local administration of vancomycin powder may produce highly  intra-wound concentrations preventing the onset of wound infection/PJI while yielding low systemic levels to prevent infections. [The last part in bold is unclear. Are the authors comparing intra-wound to oral administration of antibiotic? If so please rewrite the sentence to clearly define Johnson's findings. 

Comments on the Quality of English Language

See comment above.
